

**Non-negligible secondary contribution to brown carbon in autumn and winter:**
**inspiration from particulate nitrated and oxygenated aromatic compounds in**
**urban Beijing**

Yanqin Ren[1], Zhenhai Wu[1], Yuanyuan Ji[1], Fang Bi[1], Junling Li[1], Haijie Zhang[1], Hao
Zhang [1], Hong Li[1*], Gehui Wang[2*]
[1] State Key Laboratory of Environmental Criteria and Risk Assessment, Chinese
Research Academy of Environmental Sciences, Beijing 100012, China
[2] Key Lab of Geographic Information Science of Ministry of Education of China,
School of Geographic Sciences, East China Normal University, Shanghai 200142,
China
*Corresponding authors: Dr. Gehui Wang/ Dr. Hong Li
E-mail address: ghwang@geo.ecnu.edu.cn / lihong@craes.org.cn



**Abstract**
Nitrated aromatic compounds (NACs) and oxygenated derivatives of polycyclic
aromatic hydrocarbons (OPAHs) play vital roles within brown carbon (BrC),
influencing both climate dynamics and human health to a certain degree. The
concentrations of these drug classes were analyzed in $PM_{2.5}$ from an urban area in
Beijing during the autumn and winter of 2017–2018. There were four heavy haze
episodes during the campaign; two of which happened prior to heating whereas the
other two after heating. During the entire course of sampling, the mean total
concentrations of the nine NACs and the eight OPAHs were 1.2-263 and 2.1-234 ng m$^-$
$^3$, respectively. The concentrations of both NACs and OPAHs were approximately 2-3
times higher in the heating period than before heating. For NACs, the relative molecular
composition did not change significantly before and during heating, with 4-
nitrocatechol and 4-nitrophenol demonstrating the highest abundance. For OPAHs, 1-
Naphthaldehyde was the most abundant species before and during heating, while the
relative proportion of Anthraquinone increased by more than twice, from 13% before
heating to 31% during the heating. In Beijing's urban area during autumn and winter,
significant sources of NACs and OPAHs have been traced back to automobile emissions
and biomass-burning activities. Interestingly, it was observed that the contribution from
coal combustion increased notably with the onset of heating during this period. It is
worth noticing that the secondary generation of BrC was important throughout the
whole sampling period, which was manifested by the photochemical reaction before



heating and the aqueous reaction during heating. It was further found that the haze in
autumn and winter was nitrate-driven before heating and SOC-driven when heating
began, and the secondary formation of BrC increased significantly in pollution events,
particularly during heating.
**1 Introduction**
As an important light-absorbing material, brown carbon (BrC) has garnered
increasing attention in recent years (Jiang et al., 2023; Yi Chen et al., 2022; Song et al.,
2022; Cai et al., 2022; Zhang et al., 2021; Liu et al., 2023; Ren et al., 2023; Ren et al.,
2022). BrC could not only directly absorb solar energy, but also indirectly contribute to
climate change by promoting the evaporation of water and the dispersal of clouds
(Laskin et al., 2015; Huang et al., 2018). In addition to its significant climate effects,
BrC also has potential adverse effects on human health on account of its strong
mutagenic, cytotoxic, and carcinogenic properties (Teich et al., 2016).
Primary as well as secondary sources contribute to the atmospheric accumulation
of BrC (Zhu et al., 2021). Direct emissions of primary BrC come from burning biomass
and combustion of fossil fuels (Ni et al., 2021; Wang et al., 2020a; Lu et al., 2019a; Lu
et al., 2019b). Oxidation and aging processes in the atmosphere produce secondary BrC
(Wang et al., 2019; Wang et al., 2020c; Cheng et al., 2021; Jiang et al., 2023; Cai et al.,
2022). Toluene, phenol, benzene, and other aromatic hydrocarbons can be oxidized by
$NO_3$ or OH radical vapor phase in the presence of $NO_x$ to produce nitrophenol or



nitrocatechol (Olariu et al., 2002; Sato et al., 2007; Iinuma et al., 2010; Ji et al., 2017).
VOCs emitted during biomass combustion and pyrolysis (such as cresol, catechol,
methyl catechol, etc.) can be oxidized to produce nitro-aromatic hydrocarbons (Iinuma
et al., 2010; Claeys et al., 2012; Finewax et al., 2018). Research on the source analysis
of brown carbon (BrC) frequently focuses on examining two key constituents: the
carbon component within humic-like substances (HULIS-C) and water-soluble organic
carbon (WSOC). These components are often studied to understand the origins and
properties of BrC in various environmental contexts. Secondary generation and burning
of biomass are the two main sources of HULIS in Guangzhou and Shanghai (Fan et al.,
2016; Zhao et al., 2016). In comparison with the water-insoluble BrC in the winter, the
contribution of non-fossil sources (for instance burning biomass) to water-soluble BrC
is 67% (Liu et al., 2018; Song et al., 2018). Coal combustion is presumably a significant
source of HULIS in the winter, in addition to burning biomass and secondary generation
(Tan et al., 2016). According to multiple studies conducted in Beijing, the primary
contributor to WSOC is secondary generation, accounting for 54% of its composition.
Following this, biomass burning contributes approximately 40%, while other primary
emission sources contribute a smaller proportion, making up only 6% (Du et al., 2014).
In Beijing, the percentages of biomass burning, coal combustion, and secondary
generation that contribute to atmospheric HULIS are 47%, 15%, and 39%, respectively.
The primary origins of HULIS show minimal association with motor vehicles and
industrial emissions (Li et al., 2019). According to Ma et al., secondary generation is
responsible for over 50% of HULIS in the non-heating season. Biomass burning



represents 21% of the HULIS content during this period. However, in the heating
season, approximately 40% of HULIS originates from biomass burning, while the
remaining 60% is contributed by diverse combustion sources like coal burning, waste
incineration, and vehicular emissions. Within this season, secondary generation
accounts for about 19% of the HULIS content (Ma et al., 2018).
The research suggests that various sources contribute to BrC, but their relative
impact varies depending on time and location. As a result, the chemical makeup, light
absorption characteristics, and concentrations of BrC show considerable variability.
This variability poses challenges in accurately assessing and forecasting the influence
of these sources on radiation and climate changes (Wang et al., 2020b; Yan et al., 2018;
Laskin et al., 2015). However, until recently, there was only a limited volume of
research pertaining to the sources and pathways of BrC leading to their generation in
the densely populated city environment. NACs and OPAHs are the primary focus of
this study because several studies have noted that nitrogen-containing aromatics,
polycyclic aromatic hydrocarbons (PAHs), and their derivatives are significant BrC
chromophores (Huang et al., 2018; Cai et al., 2022; Yi Chen et al., 2022; Wu et al.,
2020; Liu et al., 2023; Wang et al., 2020b). It is well-established that residential heating
plays a significant role in the substantial increase of anthropogenic pollutant emissions
during the winter season. From autumn to winter, there is a substantial rise in the
emission of aromatics-derived secondary organic aerosol (Ding et al., 2017), and
particle BrC is often detected especially in haze periods during the autumn and winter
(Liu et al., 2023). This study was conducted in the autumn and winter of 2017-2018 in



Beijing. Nine NACs and eight OPAHs were measured in PM$_{2.5}$ samples, with a focus
on examining their sources, compositions, and concentration variations under various
air conditions. Specifically, emphasis was placed on investigating the contribution of
secondary generation to these two typical BrC species, particularly their involvement
in particle pollution processes during autumn and winter.
**2 Materials and Methods**
**2.1 Field observations**
PM$_{2.5}$ was sampled at a height of 10m on the rooftop of a building at the Chinese
Research Academy of Environmental Sciences (CRAES), Beijing, China (40º02´N,
116º24´E). Using a high-volume sampler (1.13 m$^3$ min$^{-1}$, Thermofisher Co., USA),
PM$_{2.5}$ specimens were collected in the autumn and winter of 2017/2018. The sampling
process was executed from 8:00 to 19:30 during the day and from 20:00 to 7:30 in the
subsequent morning. The specimens and blanks were gathered using a pre-combusted
quartz fiber filter (at 450 °C for 6 h). A total of 4 field blanks and 122 PM$_{2.5}$ samples
were acquired. Individual filters were sealed in a bag made from an aluminum foil bag
before sampling and analysis and placed in a freezer set at a temperature of -20 °C.
Using automatic equipment (CRAES Supersite for Comprehensive Urban Air
Observation and Research), meteorological parameters such as air temperature (T, °C)
and relative humidity (RH, %) along with gaseous pollutants (SO$_2$, NO$_2$, O$_3$, and CO)
were observed and measured at the same time.



**2.2 Chemical analysis**
The present study employed a pre-treatment comprising ultrasonic extraction and
derivatization in an attempt to analyze the organic species in the specimens. The details
of specimen extraction and derivatization have already been published (Wang et al.,
2009; Ren et al., 2021; Ren et al., 2023). In brief, filter aliquots were sectioned and
extracted with a methanol and dichloromethane (1:2 v/v,) mixture. Following the
concentration of the extracts to dryness, derivatization was carried out using a mixture
of N, O-bis-(trimethylsilyl) trifluoroacetamide [BSTFA+TMCS, (99:1), v/v] and
pyridine (5:1, v/v). Lastly, the derivatized samples were examined using gas
chromatography coupled with a mass spectroscopy detector (GC/MS: HP 7890A, HP
5975C, Agilent Co., USA). The extraction and derivatization methods described above
allowed for the simultaneous measurement of the samples' polar and non-polar
constituents.
Given that OPAHs and NACs were the main points of focus, this study
investigated a total of eight OPAHs and nine NACs. The nine NACs included 2,4-
dinitrophenol (2, 4-DNP), 4-nitrophenol (4NP), 3-methyl-4-nitrophenol (3M4NP), 4-
nitrocatechol (4NC), 4-methyl-5-nitrocatechol (4M5NC), 4-nitroguaiacol (4NGA), 5-
nitroguaiacol (5NGA), 3-nitro-salicylic acid (3NSA), and 5-nitro-salicylic acid (5NSA),
while the eight OPAHs encompassed 9-fluorenone (9-FO), benzanthrone (BZA), 1-
Naphthaldehyde (1-NapA), anthraquinone (ATQ), 1,4-chrysenequione (1,4-CQ),
benzo(a)anthracene-7,12-dione (7,12-BaAQ), 5,12-naphthacenequione (5,12-NAQ)
and 6H-benzo(cd)pyrene-6-one (BPYRone).



The elemental carbon (EC) and organic carbon (OC) content of individual $PM_{2.5}$
filter samples were analyzed using an Atmoslytic Inc. DRI model 2001 Carbon
Analyzer. This analysis followed the Interagency Monitoring of Protected Visual
Environments (IMPROVE) thermal/optical reflectance (TOR) protocol, involving the
examination of a 0.526 $cm^2$ punch from each specimen. The specifics of the above-
described techniques have been documented in literature (Li et al., 2016; Ren et al.,

2021).

**2.3 Evaluation of secondary BrC**
In this study, the contributions of secondary oxidation to the detected NACs and
OPAHs were evaluated by using a CO-tracer method, which is comparable to the EC-
tracer used for secondary OC quantification. Various methodologies have been
similarly adopted successfully in other studies (Liu et al., 2023; Cai et al., 2022).
Equation 1 and Equation 2 were respectively used to evaluate the secondary formation
of NACs and OPAHs.
$$[NACs]_s = [NACs]_t - ([NACs]_t/[CO])_{pri} \times [CO] \quad\quad (1)$$
$$[OPAHs]_s = [OPAHs]_t - ([OPAHs]_t/[CO])_{pri} \times [CO] \quad\quad (2)$$
$[NACs]_s$ and $[NACs]_t$ in Equation 1 refer to the NACs concentration produced by
secondary oxidation and the total amount of NACs, respectively. $([NACs]_t/[CO])_{pri}$
represents the primary emission ratio of NACs in relation to combustion. This
calculation assumes that the primary source was predominant during the period, with
minimal secondary production. The $([NACs]_t/[CO])_{pri}$ was calculated in this work by
fitting the 15% lowest $[NACs]_t/[CO]$ ratios observed during the entire sampling





duration. In equation 2 the concentration of OPAHs produced by secondary oxidation
and the total observed OPAHs are denoted by [OPAHs]$_s$ and [OPAHs]$_t$ respectively.
The concentration of CO is denoted by [CO], while the primary emission ratio of
OPAHs in relation to combustion is represented by ([OPAHs] /[CO]) $_{pri}$, which was
calculated by fitting the lowest 15% [OPAHs]$_t$ /[CO] ratios observed in the entire
sample interval.
**3 Results and discussion**
**3.1 Variations in major components of PM$_{2.5}$ with respect to meteorological**
**conditions and gaseous pollution**
Based on the Beijing heating time, the entire period of the study was divided into
two phases: before heating (18 October to 14 November 2017) and during heating (15
to 23 November 2017; 23 December 2017 to 17 January 2018). Table 1 and Fig. 1
present the temporal fluctuations in meteorological factors, gaseous pollutant
concentrations, and the main PM$_{2.5}$ components in the two phases. The temperature (T)
and relative humidity (RH) were higher before heating (11 ± 3.8 ℃ and 49 ± 26%) than
during heating (1.9 ± 4.4 ℃ and 23 ± 15%), with average values amounting to 5.9 ±
5.9 ℃ and 35 ± 25%, respectively. SO$_2$ concentrations during heating (4.3 ± 1.5 ppb)
were more than twice that before heating (2.1 ± 0.8 ppb), presumably because of the
increase in household coal burning for heating. The levels of NO$_2$ and NO remained
consistent before and during heating, suggesting that these pollutants were minimally



impacted by heating and were primarily influenced by mobile sources in Beijing. This
pattern seems to remain stable in the short term.

Fig.2 shows the variation in the chemical makeup of $PM_{2.5}$ in the entire sampling

period, before and during hearting, respectively. Secondary inorganic aerosols (SIA, i.e.
$SO_4^{2-}$, $NH_4^+$, and $NO_3^-$) were identified as the leading constituents of $PM_{2.5}$, followed
by OM (1.6 times OC), with an average of 31.5% and 20.4% in the whole sampling,
respectively (Fig. 2a). Even though the $PM_{2.5}$ concentrations remained relatively stable
during this period (as indicated in Table 1 and Fig. 2), there were significant changes
observed in the concentrations of SIA and OM, as well as their relative contributions to
$PM_{2.5}$. Before heating, SIA accounted for 41.9% of $PM_{2.5}$, which notably decreased to
23.1% during heating. This decline was primarily evident in the reduction of $NO^{3-}$.
Specifically, $SO_4^{2-}$, $NO_3^-$, and $NH_4^+$ were measured at 5.5, 16, and 5.4 μg m$^{-3}$,
respectively, according to Table 1. These values constituted 8.7%, 24.7%, and 8.5% of
$PM_{2.5}$ before heating, as shown in Fig. 2b. Their concentrations decreased to 4.3, 6.8,
and 4.2 μg m$^{-3}$ (Table 1). The relative contributions of $NO_3^-$ to $PM_{2.5}$ dropped
dramatically to 10.3% during heating, amounting to a drop of nearly 60%. Both $SO_4^{2-}$
and $NH_4^+$ experienced a roughly 25% decrease in their relative contributions to $PM_{2.5}$,
as illustrated in Fig. 2c. The relative abundance of OM to $PM_{2.5}$ increased from 18.6%
before heating to 21.9% during heating, with the average mass concentration of OC
showing an increase from 7.4 to 9.1 μg m$^{-3}$ in the corresponding duration. The OC/EC
ratio also increased by 63% from 2.7 ± 3.3 before heating to 4.4 ± 3.7 during heating.
These significant changes in SIA and OM, including concentrations and the relative



contributions to PM₂.₅, showed that primary organic aerosols and/or VOCs emissions
were the leading contributors during the heating seasons due to household heating (Tan
et al., 2018). The rise in mass concentrations of $K^+$ and $Cl^-$ indicated additional burning
activities occurring during heating, aside from coal combustion, such as biomass
burning (Bai et al., 2023; Li et al., 2022).

Table 1. Gaseous pollution concentrations and meteorological parameters and chemical constituents of PM₂.₅ during the sampling periods in Beijing.

| | The whole sampling $N$=122 | Before heating period 18/10–14/11, 2017 $N$=56 | During heating period 15/11–23/11, 2017 23/12, 2017–17/1, 2018 $N$=66 |
|---|---|---|---|
| **Meteorological parameters** | | | |
| Temperature, ℃ | 5.9 ± 5.9 ((-7.5) – 16) | 11 ± 3.8 (1.2 – 16) | 1.9 ± 4.4 ((-7.5) – 11) |
| Relative humidity, % | 35 ± 25 (7.1 – 99) | 49 ± 26 (11 – 99) | 23 ± 15 (7.1 – 67) |
| **Gaseous pollutants, ppb** | | | |
| $SO_2$ | 3.2 ± 1.6 (1.1 – 7.9) | 2.1 ± 0.8 (1.1 – 4.8) | 4.3 ± 1.5 (2.2 – 7.9) |
| $NO_2$ | 26 ± 13 (4.6 – 56) | 25 ± 11 (4.6 – 43) | 26 ± 14 (5.5 – 56) |
| NO | 26 ± 28 (2.4 – 136) | 28 ± 30 (2.4 – 136) | 25 ± 26 (2.7 – 116) |
| CO | 0.64 ± 0.55 (0.03 – 2.7) | 0.81 ± 0.42 (0.12 – 1.6) | 0.50 ± 0.61 (0.03 – 2.7) |
| **Major components of PM₂.₅, μg m⁻³** | | | |
| PM₂.₅ | 65 ± 40 (6.1 – 195) | 64 ± 39 (6.1 – 175) | 66 ± 41 (8.6 – 195) |
| OC | 8.3 ± 5.0 (0.99 – 26) | 7.4 ± 3.9 (1.0 – 18) | 9.1 ± 5.8 (1.8 – 26) |
| EC | 4.7 ± 4.7 (0.11 – 25) | 4.9 ± 3.8 (0.11 – 17) | 4.5 ± 5.3 (0.18 – 25) |
| OC/EC | 3.7 ± 3.6 (0.96 – 21) | 2.7 ± 3.3 (0.96 – 21) | 4.4 ± 3.7 (1.0 – 17) |
| $SO_4^{2-}$ | 4.8 ± 4.2 (0.85 – 25) | 5.5 ± 3.5 (0.86 – 13) | 4.3 ± 4.7 (0.85 – 25) |
| $NO_3^-$ | 11 ± 14 (0.09 – 58) | 16 ± 16 (0.09 – 58) | 6.8 ± 8.8 (0.29 – 37) |
| $NH_4^+$ | 4.7 ± 4.9 (0.02 – 20) | 5.4 ± 5.4 (0.02 – 20) | 4.2 ± 4.5 (0.19 – 20) |
| $K^+$ | 0.43 ± 0.39 (0.02 – 2.2) | 0.38 ± 0.27 (0.03 – 1.1) | 0.48 ± 0.46 (0.02 – 2.2) |
| $Cl^-$ | 1.5 ± 1.6 (0.06 – 9.2) | 1.0 ± 0.98 (0.06 – 4.5) | 1.9 ± 2.0 (0.13 – 9.2) |







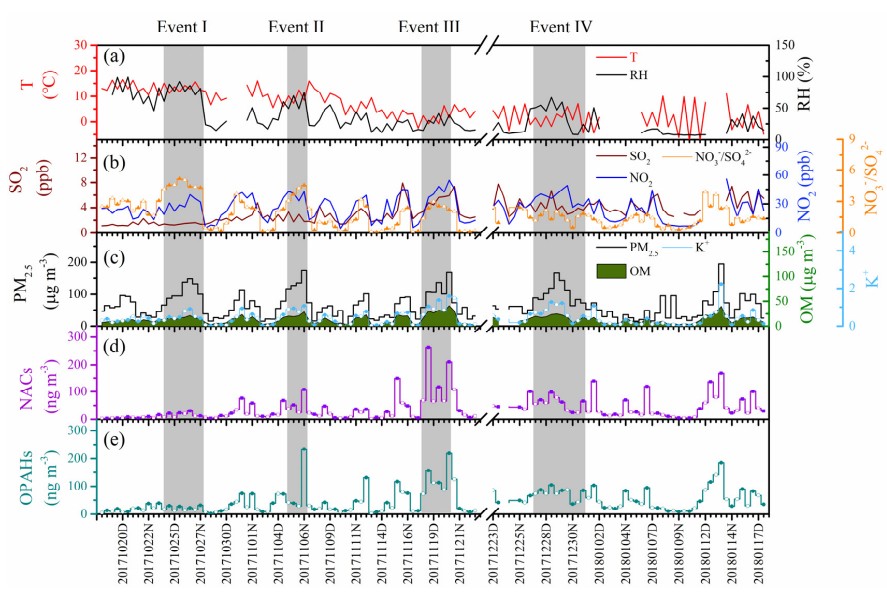

Fig.1 Time series of (a) RH and T, (b) $SO_2$ and $NO_2$, (c) $PM_{2.5}$, OM, and $K^+$, (d) NACs and (e) OPAHs in the autumn and winter of urban Beijing. (Daytime is denoted by empty marks and the nighttime is represented by solid marks in the panel b–e. The pollution episodes, with elevated concentrations of daily $PM_{2.5}$ more than 75μg m$^{-3}$ in two successive days, have been marked in light gray).

Fig.2 Chemical constitution of $PM_{2.5}$ in the entire sampling period (a), before (b), and during (c) heating periods, respectively.

**3.2 Concentration and composition variations of BrC compounds**

This work quantified nine NACs and eight OPAHs. The corresponding





concentrations and compositions have been presented in Fig. 3 and Table S1
(supporting information).

As seen in Table S1, during the entire sampling, the total concentrations of NACs

($\sum$9NACs) and their corresponding contribution to OM ($\sum$9NACs/OM) respectively
averaged to 38 (1.2–263) ng m$^{-3}$ and 0.25 (0.03–0.86) %. $\sum$9NACs and $\sum$9NACs/OM
respectively averaged 53 (4.5–263) ng m$^{-3}$ and 0.33 (0.09–0.86) % during heating, both
values are two times higher in magnitude in comparison to those measured before
heating (averaged 20 (1.2–108) ng m$^{-3}$ and 0.15 (0.03–0.4) %, respectively). $\sum$9NACs
exhibited a nighttime increase, reaching approximately twice the levels observed during
daytime throughout the entire campaign (Fig. 3a). The observed difference between day
and night is consistent with our previous research (Ren et al., 2022). However, the
relative molecular composition of the total nine NACs in PM$_{2.5}$ did not manifest any
significant change (Fig.4a, c), 4-Nitrophenol (4NP) was found to have the highest
concentration among all species, accounting for 44% and 42% of the total NACs before
and during heating, followed by 4-nitrocatechol (4NC) which accounted for 21% before
heating and 23% during heating. These findings align with the dominant species
observed in previous studies (Ren et al., 2022; Ren et al., 2023; Li et al., 2020) however
the values were much higher in comparison to those found in our earlier work (Ren et
al., 2022) at the same sample site during the spring (8.6 (0.48–27) ng m$^{-3}$) and summer
(8.5 (1.0–16) ng m$^{-3}$). It's plausible that seasonal variations in NACs are linked to
emission sources, formation pathways, and weather conditions. In this study, the overall
abundance of the $\sum$9NACs appeared to align closely with measurements from earlier



studies conducted during winter in Beijing (74 ± 51 ng m$^{-3}$ in winter, 20 ± 12 ng m$^{-3}$ in
autumn) (Li et al., 2020) and Jinan (48 ± 26 ng m$^{-3}$ in winter, 9.8 ± 4.2 ng m$^{-3}$ in autumn,)
(Wang et al., 2018), but are significantly higher than those measured for Xi'an (17 ± 12
ng m$^{-3}$) and Hong Kong (12 ± 14 ng m$^{-3}$) in winter (Wu et al., 2020; Chow et al., 2015).
In contrast to studies conducted abroad, the levels of ∑9NACs in this particular study
tended to be higher Germany showed 16 ng m$^{-3}$, while in the UK, levels were around
19 ng m$^{-3}$. Belgium recorded levels of 32 ng m$^{-3}$ in winter and 13 ng m$^{-3}$ in autumn.
(Teich et al., 2017; Mohr et al., 2013; Kahnt et al., 2013). This indicates that it is urgent
to further reduce the concentration of contaminant precursors in China.

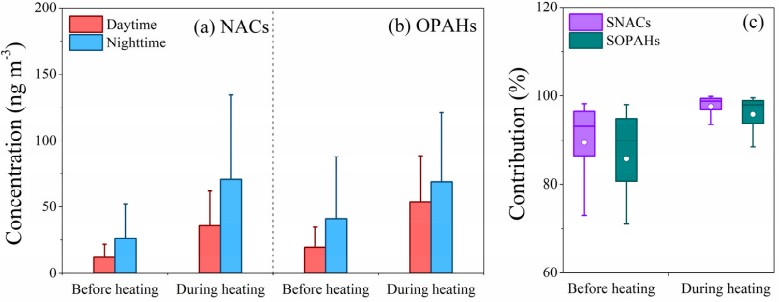


Fig.3 NACs and OPAHs concentrations (a,b) and contributions of secondary
formation (SNACs, SOPAHs) to the total (c) before and during heating periods.
Throughout the sampling, the total concentrations of OPAHs (∑8OPAHs)
averaged 47 (1.2–234) ng m$^{-3}$ whereas the mean value for their total contribution to OM
(∑8OPAHs/OM) was 0.33 (0.06–0.81) % (Table S1). These values were both slightly
higher than those of NACs in this work. ∑8OPAHs and ∑8OPAHs/OM respectively
averaged 61 (6.9–218) ng m$^{-3}$ and 0.40 (0.18–0.58) % during heating. These values are
almost twice as much as those measured before heating (averaging 31 (2.1–234) ng m$^{-}$
$^{3}$ and 0.24 (0.06–0.81) %, respectively). Like the ∑9NACs, the combined levels of

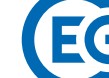

∑8OPAHs were higher during nighttime compared to daytime, averaging about twice
as high before heating and 1.3 times during heating, as indicated in Fig. 3b. Among the
eight OPAHs studied, 1-NapA constituted the highest proportion before (60%) and
during (36%) heating. However, the relative proportion of ATQ more than doubled,
increasing from 13% before heating to 31% during heating, as depicted in Fig. 4b and
d. The average concentrations of ∑8OPAHs were higher than those recorded for other
Chinese urban sites, including Guangzhou (23 ng m$^{-3}$) and Xi'an (54 ng m$^{-3}$) (Ren et
al., 2017) as well as higher than those documented for the south (41.8 ng m$^{-3}$, traffic
site) (Alves et al., 2017) and central (~10 ng m$^{-3}$) European cities (Lammel et al., 2020).
The average concentrations of ∑8OPAHs were also higher than those recorded for
Mainz, Germany (0.047-1.6 ng m$^{-3}$) and Thessaloniki, Greece (0.86-4.3 ng m$^{-3}$)
(Kitanovski et al., 2020).

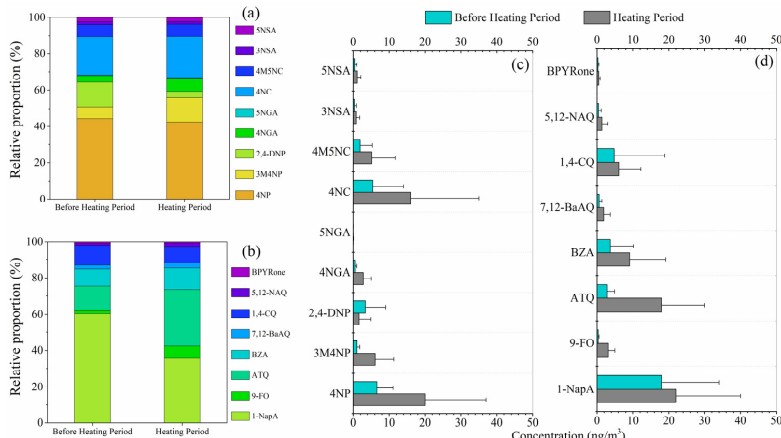


Fig.4 Comparison of measurements before and during the heating period at the urban
site of Beijing, including (a) Relative proportion of NAC species, (b) Relative
proportion of OPAH species, (c) NAC concentrations, and (d) OPAH concentrations.
( 4NP: 4-nitrophenol, 3M4NP: 3-methyl-4-nitrophenol, 2, 4-DNP: 2,4-dinitrophenol,
4NGA: 4-nitroguaiacol, 5NGA: 5-nitroguaiacol, 4NC: 4-nitrocatechol, 4M5NC: 4-





methyl-5-nitrocatechol, 3NSA: 3-nitro-salicylic acid, 5NSA: 5-nitro-salicylic acid; 1-
NapA: 1-Naphthaldehyde, 9-FO: 9-fluorenone, ATQ: anthraquinone, BZA:
benzanthrone, 7,12-BaAQ: benzo(a)anthracene-7,12-dione, 1,4-CQ: 1,4-
chrysenequione, 5,12-NAQ: 5,12-naphthacenequione, and BPYRone: 6H-
benzo(cd)pyrene-6-one)

**3.3 Sources and formation of BrC compounds**

The relation between individual and total species and the associated pollutants—
levoglucosan, $K^+$, $SO_2$, $NO_2$, $O_3$, RH, and SIA—was examined according to the data
findings for the Pearson correlations shown in Table 2 (for NACs) and Table 3 (for
OPAHs) to provide additional clarity regarding the source and formation of NACs and
OPAHs. The strong correlations observed throughout the entire campaign between
levoglucosan (an organic tracer associated with biomass burning), $K^+$ (an inorganic
tracer linked to biomass burning), and $NO_2$ with total NACs and all identified NAC
species indicate that both automobile emissions and biomass burning played significant
roles in the accumulation of NACs in urban Beijing during autumn and winter. The
correlation between NACs and $SO_2$ was significantly higher during heating (r=0.275,
p<0.05) compared to pre-heating, indicating that coal combustions play a more
significant role in NAC formation after heating commences.
In addition to these primary pollutants, NACs were also significantly correlated
with some secondary pollutants. Before heating, there existed a strong positive
association (r=0.692, p<0.01) between NACs and $O_3$. However, this association
changed considerably after heating, becoming notably negative (r=-0.303, p<0.05).
NACs and RH concurrently displayed a strong positive correlation (r=0.548, p<0.01)



during heating. Along with $SO_4^{2-}$, $NO_3^-$, and $NH_4^+$, total NACs also exhibited high
positive correlation, particularly while heating (r=0.373, p<0.01; r=0.504, p<0.01;
r=0.513, p<0.01, respectively). The overall concentrations of OPAHs and NACs
throughout the campaign exhibited substantial correlations (r=0.830, p<0.01 before
heating; r=0.895, p<0.01 during heating) (Table 3, Fig. S1). This suggests that their
sources and/or influencing variables were comparable. Specifically, throughout the
entire campaign, both total OPAHs and all identified OPAH species exhibited a strong
correlation with levoglucosan, $K^+$, and $NO_2$. This implies that automobile emissions
and biomass burning played significant roles as sources of OPAHs. OPAHs and $SO_2$
(r=0.365, $p$<0.01) were determined to be more strongly correlated during heating than
before heating, suggesting the contribution of coal combustions to OPAHs becomes
significant when heating begins. Moreover, the correlation between OPAHs and $O_3$ was
significantly positive before heating (r=0.563, $p$<0.01), whereas it was significantly
negative during heating (r=-0.385, $p$<0.01). Furthermore, it was discovered that
throughout the heating phase, OPAHs and RH had a substantial positive correlation
(r=0.578, $p$<0.01). Total OPAHs also showed good correlations with $SO_4^{2-}$, $NO_3^-$, and
$NH_4^+$, especially during heating period (r=0.477, $p$<0.01; r=0.658, $p$<0.01; r=0.658,
$p$<0.01; respectively).
The observed phenomena, involving photooxidation before heating and aqueous
reactions during heating, strongly suggest a significant role in the secondary creation
of BrC throughout the entire sampling period. Earlier studies have highlighted that in
certain regions, the primary mechanism driving the formation of nitro-aromatic



hydrocarbons involves the gaseous phase oxidation of VOC precursors from
anthropogenic sources, such as toluene and benzene (Olariu et al., 2002; Sato et al.,
2007; Yuan et al., 2016; Ji et al., 2017; Liu et al., 2023). According to a recent study,
for instance, NACs are mostly generated at a rural location on China's Chongming
Island through gaseous-phase photooxidation (Liu et al., 2023). Aqueous reaction is
also a key pathway for the formation of BrC (Zhang et al., 2020; Cheng et al., 2021;
Jiang et al., 2023). Zhang et al., suggested that the aqueous formation of anthropogenic
secondary organic carbon was a key source of atmospheric BrC in Xi'an (Zhang et al.,
2020). Wang et al.'s field observations in urban Beijing revealed that the aqueous
reaction is a significant mechanism for the secondary synthesis of nitro-aromatic
hydrocarbons during summer temperatures with high relative humidity (Wang et al.,

2019).


Table 2 Correlations between NACs and meteorological parameters, gas pollutants, and aerosol components before (n=56) and during the heating period (n=66).

| Before heating period | | levoglucosan | $K^+$ | $SO_2$ | $NO_2$ | $O_3$ | RH | $SO_4^{2-}$ | $NO_3^-$ | $NH_4^+$ |
|---|---|---|---|---|---|---|---|---|---|---|
| NACs | ∑9NACs | 0.897** | 0.738** | 0.210 | 0.714** | 0.692** | 0.170 | 0.359** | 0.369** | 0.190 |
| | 4NP | 0.784** | 0.699** | 0.249 | 0.715** | 0.649** | 0.118 | 0.345** | 0.372** | 0.207 |
| | 3M4NP | 0.752** | 0.526** | 0.290* | 0.575** | 0.511** | -0.011 | 0.184 | 0.191 | 0.029 |
| | 2,4-DNP | 0.436** | 0.353** | 0.310* | 0.463** | 0.492** | -0.151 | 0.034 | -0.048 | -0.166 |
| | 4NGA | 0.545** | 0.361** | 0.438** | 0.560** | 0.524** | -0.137 | -0.016 | -0.025 | -0.131 |
| | 5NGA | 0.582** | 0.355** | 0.114 | 0.343** | 0.433** | 0.005 | 0.120 | 0.139 | 0.044 |
| | 4NC | 0.897** | 0.748** | 0.064 | 0.641** | 0.617** | 0.308* | 0.448** | 0.486** | 0.325* |
| | 4M5NC | 0.885** | 0.668** | 0.076 | 0.579** | 0.577** | 0.252 | 0.364** | 0.413** | 0.261 |
| | 3NSA | 0.791** | 0.678** | 0.129 | 0.553** | 0.495** | 0.214 | 0.457** | 0.515** | 0.331* |
| | 5NSA | 0.737** | 0.596** | 0.219 | 0.594** | 0.553** | 0.085 | 0.279* | 0.316* | 0.125 |
| During heating period | | levoglucosan | $K^+$ | $SO_2$ | $NO_2$ | $O_3$ | RH | $SO_4^{2-}$ | $NO_3^-$ | $NH_4^+$ |
| NACs | ∑9NACs | 0.888** | 0.786** | 0.275* | 0.481** | -0.303* | 0.548** | 0.373** | 0.504** | 0.513** |
| | 4NP | 0.812** | 0.725** | 0.262* | 0.471** | -0.296* | 0.586** | 0.390** | 0.489** | 0.511** |
| | 3M4NP | 0.756** | 0.655** | 0.248 | 0.374** | -0.225 | 0.613** | 0.318** | 0.397** | 0.462** |



| | | | | | | | | | |
|---|---|---|---|---|---|---|---|---|---|
| 2,4-DNP | 0.537** | 0.495** | 0.280* | 0.417** | -0.304* | 0.199 | 0.136 | 0.247* | 0.136 |
| 4NGA | 0.672** | 0.406** | 0.229 | 0.274* | -0.206 | 0.201 | -0.047 | 0.074 | 0.081 |
| 5NGA | 0.275* | 0.208 | -0.028 | 0.190 | 0.026 | -0.006 | 0.114 | 0.100 | 0.125 |
| 4NC | 0.894** | 0.804** | 0.248 | 0.454** | -0.290* | 0.520** | 0.378** | 0.523** | 0.530** |
| 4M5NC | 0.882** | 0.736** | 0.246 | 0.434** | -0.244 | 0.430** | 0.283* | 0.421** | 0.422** |
| 3NSA | 0.788** | 0.910** | 0.348** | 0.681** | -0.410** | 0.577** | 0.707** | 0.888** | 0.828** |
| 5NSA | 0.820** | 0.866** | 0.268* | 0.629** | -0.377** | 0.599** | 0.680** | 0.846** | 0.828** |

**significant correlation at the 0.01 level;

*significant correlation at the 0.05 level;

From the above analysis, it is evident that there is a good correlation between these
two aromatic compounds and levoglucosan, as well as $SO_2$. Since levoglucosan and
$SO_2$ are long-lived and inert chemicals in the atmosphere (Cai et al., 2022), it was not
possible to determine with certainty whether NACs and OPAHs originated
predominantly from direct emission from the coal and biomass combustion or by
secondary oxidation of the precursors produced as a result of these processes. Equations
1 and 2's outcomes indicated that in Beijing's urban areas during fall and winter, NACs
and OPAHs were predominantly of secondary origin. Throughout the entire sampling
period, secondary formation accounted for 17% to 99% (average of 80%) of NACs and
8.9% to 99% (average of 73%) of OPAHs, as depicted in Fig. 3c. Notably, the secondary
fraction for OPAHs increased by 10.4% from 86% to 96%, while the secondary fraction
for NACs rose by 8.9% from 90% before heating to 98% during heating. Earlier studies
have highlighted the presence of significant levels of secondary particle BrC during
autumn and winter, particularly during haze periods (Ding et al., 2017; Liu et al., 2023),
and the results of this work corroborate well with the earlier studies. Moreover, the good
correlations between OPAHs and NACs with $O_3$ before heating and with RH during
heating, confirm the importance of photochemical and aqueous oxidation in these two



different periods.

Table 3 Correlations between OPAHs and meteorological parameters, gas pollutants, and aerosol components before (n=56) and during the heating period (n=66).

| Before heating period | | ∑9NACs | levoglucosan | $K^+$ | $SO_2$ | $NO_2$ | $O_3$ | RH | $SO_4^{2-}$ | $NO_3^-$ | $NH_4^+$ |
|---|---|---|---|---|---|---|---|---|---|---|---|
| OPAHs | ∑8OPAHs | 0.830** | 0.865** | .605** | 0.188 | 0.563** | 0.563** | 0.143 | 0.244 | 0.283* | 0.139 |
| | 1-NapA | 0.844** | 0.870** | .621** | 0.211 | 0.640** | 0.622** | 0.174 | 0.213 | 0.238 | 0.096 |
| | 9-FO | 0.775** | 0.785** | .646** | 0.283* | 0.558** | 0.573** | 0.059 | 0.183 | 0.235 | 0.119 |
| | ATQ | 0.633** | 0.694** | .497** | 0.392** | 0.477** | 0.483** | -0.018 | 0.042 | 0.061 | -0.024 |
| | BZA | 0.686** | 0.759** | .573** | 0.232 | 0.537** | 0.594** | 0.110 | 0.177 | 0.206 | 0.117 |
| | 7,12-BaAQ | 0.821** | 0.865** | .685** | 0.189 | 0.591** | 0.622** | 0.187 | 0.325* | 0.356** | 0.224 |
| | 1,4-CQ | 0.636** | 0.646** | .406** | 0.041 | 0.303* | 0.290* | 0.102 | 0.259 | 0.309* | 0.174 |
| | 5,12-NAQ | 0.694** | 0.752** | .563** | 0.227 | 0.511** | 0.559** | 0.091 | 0.184 | 0.214 | 0.110 |
| | BPYRone | 0.827** | 0.870** | .662** | 0.131 | 0.590** | 0.616** | 0.226 | 0.345** | 0.398** | 0.255 |
| Heating period | | ∑9NACs | levoglucosan | $K^+$ | $SO_2$ | $NO_2$ | $O_3$ | RH | $SO_4^{2-}$ | $NO_3^-$ | $NH_4^+$ |
| OPAHs | ∑8OPAHs | 0.895** | 0.931** | 0.877** | 0.365** | 0.678** | -0.385** | 0.578** | 0.477** | 0.658** | 0.658** |
| | 1-NapA | 0.752** | 0.774** | 0.659** | 0.248 | 0.547** | -0.378** | 0.332* | 0.332** | 0.498** | 0.446** |
| | 9-FO | 0.478** | 0.457** | 0.342** | 0.305* | 0.302* | -0.020 | 0.131 | -0.013 | 0.167 | 0.250* |
| | ATQ | 0.780** | 0.797** | 0.815** | 0.426** | 0.668** | -0.372** | 0.656** | 0.466** | 0.642** | 0.684** |
| | BZA | 0.840** | 0.881** | 0.829** | 0.332** | 0.570** | -0.273* | 0.577** | 0.431** | 0.568** | 0.577** |
| | 7,12-BaAQ | 0.801** | 0.856** | 0.896** | 0.391** | 0.633** | -0.299* | 0.655** | 0.531** | 0.689** | 0.710** |
| | 1,4-CQ | 0.703** | 0.780** | 0.791** | 0.244 | 0.597** | -0.282* | 0.624** | 0.560** | 0.647** | 0.675** |
| | 5,12-NAQ | 0.777** | 0.818** | 0.869** | 0.365** | 0.601** | -0.293* | 0.535** | 0.444** | 0.619** | 0.614** |
| | BPYRone | 0.858** | 0.857** | 0.845** | 0.339** | 0.588** | -0.320* | 0.478** | 0.396** | 0.612** | 0.568** |

**significant correlation at the 0.01 level;

*significant correlation at the 0.05 level;


## 3.4 Different pollution characteristics in haze events

From Fig.1, it can be found that $PM_{2.5}$ shows four equivalent maxima lasting for
two to five days. Among the four pollution events, two occurred before heating (24-27,
October and 5-6, November) and the other two occurred during heating (18-20,
November and 27-31 December). $PM_{2.5}$ was significantly different in terms of its
chemical constituents before and during heating although the mass concentration of



PM$_{2.5}$ was rather similar (respectively averaging 107, 115, 113, and 108 μg m$^{-3}$ for
Event I, II, III, and IV) (Fig. 5a). In the two events before heating, OM existed as the
second most dominant species in PM$_{2.5}$, with the respective relative abundance of 15.5%
and 16.2% in Events I and II. In contrast, OM surfaced as the most dominant species of
PM$_{2.5}$ during heating. The relative abundance of OM (26.5%) during Event III was
higher (17.8%) than that during Event IV (Fig.5a). The ratios of NO$_3^-$/PM$_{2.5}$ were higher
in Events I and II as compared to Events III and IV, with the ratios of SOC/OC showing
the opposite trend (Fig.5b), suggesting a significant increase in the concentration of
secondary organic compounds after heating. In the context of fossil fuel combustion,
PAHs serve as markers for coal burning, while levoglucosan acts as a significant tracer
for biomass smoke. Figure 5c shows that the ratios of PAHs to organic carbon mass in
PM$_{2.5}$ (PAHs/OC) were higher during Events III and IV compared to Events I and II.
This underscores the heightened emissions from household burning of coal for heating
purposes. Levoglucosan/OC, the mass ratio of levoglucosan to OC in PM$_{2.5}$, did not,
however, rise considerably over the same period (Fig. 5c), indicating a similar degree
of emissions from burning biomass before and during heating. This result was
consistent with our earlier research from the 2014 APEC meeting (Wang et al., 2017).



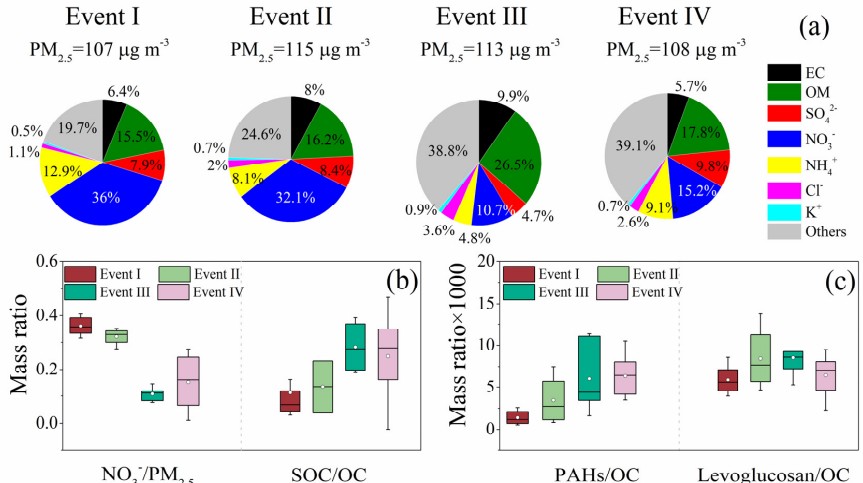

Fig.5 Comparative analysis of the chemical composition of PM$_{2.5}$ during four distinct events of air pollution. (a) Relative percentages of major species in PM$_{2.5}$; (b, c) mass ratios of the key species and organic tracers in PM$_{2.5}$.

According to the majority of research, Beijing's haze is distinguished by intense secondary formation (Zhang et al., 2018; Xu et al., 2017; Sun et al., 2016; Guo et al., 2014). According to several research studies, organic materials (OM) predominates in the autumn and winter, while secondary SIA is the most prevalent species in the summer (Renhe et al., 2014). Additionally, according to a few investigators, SIA has a major role in wintertime pollution episodes (Guo et al., 2014; Wang et al., 2016). Furthermore, a recent investigation identified the species responsible for Beijing haze, and listed distinct haze-driving species operative over the year: The haze is primarily OM-driven during winter and late fall, nitrate-driven in early fall, sulfate-driven in summer, whereas it is driven primarily by nitrates during spring (Tan et al., 2018). Table 4 and Fig.5a depict that PM$_{2.5}$ was enriched with SIA especially NO$_3^-$ during Events I and II, but enriched OM with higher levels of SOC was observed during Events III and IV. The findings strongly indicated that haze during fall and winter in urban Beijing





was primarily influenced by nitrate before heating and shifted to being driven by SOC
during heating. Table 4 illustrates that T (temperature) and RH (relative humidity) were
notably higher during Events I and II compared to Events III and IV. These warmer and
moister conditions favored photochemical oxidation, leading to an increased abundance
of SIA during the same period. Home heating activities, such as burning residential coal,
were increased during the heating period. This resulted in massive emissions of $SO_2$,
$NO_x$, VOCs, and primary particles, all of which were conducive to the generation of
SOC. As a result, during Events III and IV, SOC concentrations and relative
abundances were higher than during Events I and II. Furthermore, Fig. 6a shows that
the NACs/OC and OPAHs/OC ratios were significantly higher in Events III and IV
compared to Events I and II. Figure 6b displays a parallel trend in the relative
contributions of secondary formation for both events, highlighting a notable increase in
the secondary formation of BrC during pollution events, particularly evident during
heating periods.

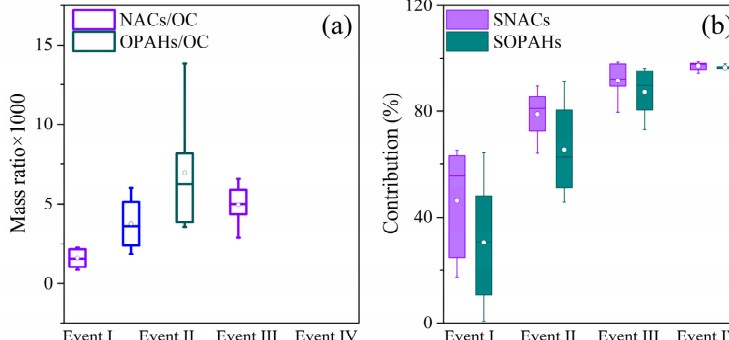


Fig. 6 Comparative analysis of the chemical composition of BrC during four distinct
air pollution events. (a) Mass ratios of NACs and OPAHs to OC in $PM_{2.5}$. (b) Relative
contributions of secondary formation (SNACs/OPAHs) to the total NACs/OPAHs in
the fine particulate.




Table 4. Meteorological parameters, chemical components (μg m$^{-3}$) of PM$_{2.5}$, and concentrations of gaseous pollutants (ppb) among four pollution episodes in Beijing.

| | Before Heating Period | | During Heating Period | |
|---|---|---|---|---|
| | Event I | Event II | Event III | Event IV |
| | 24/10–27/10 | 5/11–6/11 | 18/11–20/11 | 27/12–31/12 |
| | $N$=8 | $N$=4 | $N$=6 | $N$=10 |
| PM$_{2.5}$ | 107 ± 29 | 115 ± 48 | 113 ± 37 | 108 ± 35 |
| Temperature, ℃ | 13 ± 1.6 | 8.8 ± 4.2 | 1.8 ± 2.6 | 1.8 ± 3.8 |
| Relative humidity, % | 79 ± 10 | 55 ± 26 | 29 ± 10 | 36 ± 22 |
| SO$_2$ | 1.4 ± 0.1 | 1.9 ± 1.1 | 5.0 ± 1.0 | 4.0 ± 1.2 |
| NO | 27 ± 19 | 27 ± 15 | 51 ± 42 | 47 ± 31 |
| NO$_2$ | 30 ± 6.7 | 33 ± 14 | 40 ± 12 | 37 ± 8.1 |
| SIA [a] | 62 ± 20 | 58 ± 22 | 23 ± 9.9 | 39 ± 30 |
| NO$_3^-$ | 39 ± 13 | 38 ± 15 | 12 ± 5.2 | 18 ± 14 |
| SOC [b] | 1.9 ± 1.3 | 2.3 ± 1.8 | 7.0 ± 2.8 | 3.9 ± 2.5 |
| NACs (ng m$^{-3}$) | 17 ± 8.3 | 41 ± 36 | 131 ± 85 | 55 ± 27 |
| OPAHs (ng m$^{-3}$) | 21 ± 7.6 | 62 ± 85 | 127 ± 53 | 74 ± 23 |

[a] SIA: secondary inorganic aerosols (the sum of sulfate, nitrate, and ammonium).
[b] SOC: secondary organic carbon ([SOC]=[OC]-[EC]×([OC]/[EC])$_{pri}$). [OC]/[EC])$_{pri}$ was estimated from the fitting of the minimum [OC]/[EC] ratio, assuming that the primary source dominated the period with minimal secondary formation. In this work, ([OC]/[EC])$_{pri}$ was estimated from the fitting of the lowest 15% [OC]/[EC] ratios during the whole sampling period.

**4 Conclusions**
The current study determined the concentrations of PM$_{2.5}$-bound nine NACs and
eight OPAHs in autumn and winter in Beijing urban areas. The OPAHs and NACs
concentrations were much higher during heating than before heating. These species
have a distinct diurnal variation, with higher concentrations at night compared to day.
4-Nitrophenol, 4-nitrocatechol, and 1-Naphthaldehyde were the most abundantly
existing species in the whole campaign.
The primary sources of NACs and OPAHs were biomass combustion and
automobile emissions, with the secondary generation of BrC being the predominant



contributor across the entire sampling period. Our results underscore the significant role
of secondary generation in producing BrC, particularly its heightened contribution
during pollution events linked to heating. A comparative analysis of the chemical
constitution of $PM_{2.5}$ and BrC in four different haze events also revealed that the haze
in the autumn and winter was caused by SOC during heating and by nitrate prior to
heating. Increased attention should be directed towards reducing the emissions of
aromatic hydrocarbons and other anthropogenic volatile organic compounds (VOCs)
when heating commences. This focus is crucial for effectively mitigating pollution and
ensuring the preservation of human health. There is still only a limited volume of
research on the molecular makeup of BrC, with this study primarily concentrating on
two chromophores. As a result, further research is needed to identify more impactful
chromophores at a molecular level. Additionally, a comprehensive exploration of the
secondary generation pathways and key influencing factors of BrC through field
observations and laboratory simulations is essential. This investigation is crucial for
accurately assessing the environmental and human health impacts of BrC.
**Data availability**

The field observational and the lab experimental data used in this study are

available from the corresponding author upon request (Hong Li via
lihong@craes.org.cn).



**Author contributions**

Yanqin Ren, Gehui Wang and Hong Li designed the research; Yanqin Ren, Yuanyuan Ji and Zhenhai Wu collected the samples; Yanqin Ren, Fang Bi and Hao Zhang conducted the experiments; Yanqin Ren and Gehui Wang analyzed the data, Yanqin Ren wrote the paper; Gehui Wang, Junling Li, Haijie Zhang and Hong Li contributed to the paper with useful scientific discussions and comments.

**Competing interests**

The authors declare that they have no conflict of interest.

**Acknowledgements**

This work was supported by the Fundamental Research Funds for Central Public Welfare Scientific Research Institutes of China (No. 2022YSKY-27; No. 2019YSKY-018), and the National Natural Science Foundation of China (No. 42130704; No. 41907197).

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
