# Peer review of "Non-negligible secondary contribution to brown carbon in autumn and winter: 1 2 inspiration from particulate nitrated and oxygenated aromatic compounds in 3 urban Beijing 4 5 Yanqin Ren1, Zhenhai Wu1, Yuanyuan Ji1, Fang Bi1, Junling Li1, Haiji"

_EGUsphere, 2023_

## Referee Comment (RC2)

**The review for egusphere-2023-2593: Non-negligible secondary contribution to brown carbon in autumn and winter: inspiration from particulate nitrated and oxygenated aromatic compounds in urban Beijing**

In view of its unique light absorption properties and photochemical reactivity, brown carbon is believed to have potential impacts on global climate and atmospheric environmental quality. Therefore, brown carbon has become a hot topic in the field of environmental research in recent years. This paper uses offline methods and GC/MS to investigate the concentration variation and source characteristics of various brown carbon molecules before and during heating period in Beijing. It is found that vehicle emissions, biomass burning and coat combustion are important sources of brown carbon before and during heating period. And secondary source was important for brown carbon. However, the information of the dozen or so molecules that serve as BrC are lacking and the source apportionment are needed more logically accurate and in-depth analysis. Here are the comments.

**Major comment:**

**1.1.** The title of this article suggests that the research object is brown carbon, and the author has surfaced in the text that NAC and OPAH are two types of substances that have been confirmed as BrC molecules in previous reports. The problem is that the objects studied by the authors are actually more than a dozen confirmed molecules, some of which have been studied and confirmed as BrC molecules, determining the light absorption characteristics of these molecules, such as xie et al., 2017 and Huang, R.-J., 2018. In order to make the conclusions of the paper more rigorous, the author should supplement relevant information on the light absorption parameters of these molecules, otherwise BrC and these more than a dozen molecules are not equivalent.

**1.2.** What were the reasons behind the authors' selection of CO over levo as a tracer? In my interpretation, the long-lived and inert nature of the tracer (as per the principle and assumptions of the EC-tracer method) ensures that its mass concentration closely approximates that at the start of emission during atmospheric

transport. Consequently, the initial emission value can be derived through an emission ratio calculation, as illustrated by Eq. 1. Both levo and CO satisfy this condition, so it is reasonable to use both as tracers in the calculation. However, the authors state otherwise, citing the literature. This prompted the reviewers to question. The authors should provide a more comprehensive explanation of the fundamental assumptions and principles underlying this method, ensuring that readers can comprehend it thoroughly. The correlation analysis reveals a strong association between NAC and OPAH with levo, suggesting a preference towards biomass combustion sources. This finding contradicts the author's initial conclusion. Would the use of levo as a tracer yield a different outcome?

*Xie, M., Chen, X., Hays, M. D., Lewandowski, M., Offenberg, J., Kleindienst, T. E., and Holder, A. L.: Light Absorption of Secondary Organic Aerosol: Composition and Contribution of Nitroaromatic Compounds, Environ. Sci. Technol., 51, 11607-11616, 10.1021/acs.est.7b03263, 2017.*

*Huang, R.-J., Yang, L., Cao, J., Chen, Y., Chen, Q., Li, Y., Duan, J., Zhu, C., Dai, W., Wang, K., Lin, C., Ni, H., Corbin, J. C., Wu, Y., Zhang, R., Tie, X., Hoffmann, T., O'Dowd, C., and Dusek, U.: Brown Carbon Aerosol in Urban Xi'an, Northwest China: The Composition and Light Absorption Properties, Environ. Sci. Technol., 52, 6825-6833, 10.1021/acs.est.8b02386, 2018.*

**Other comment:**

**1.1.** Line 25: From the content of the article abstract, it seems to be expressed here during heating rather than after heating.

**1.2.** Line 68-70: 67% is a specific number, is it determined for all or for a certain site at a certain time?

**1.3.** Line 79 Please re-write citation formats correctly.

**1.4.** Line 96: Cai et al., 2022 and Yi Chen et al., 2022 did not study the importance of NAC as brown carbon but studied the NAC itself, please quote the literature correctly.

**1.5.** Line 164: Does this period indicate the whole study period or the 15% point selected by the authors? The assumption that the lowest 15% of the ratio of

NAC/CO is used to calculate emission ratios, is there any or some direct data or reports on emissions that can be used to justify that this proportion is correct to some extent?

**1.6.** Line 209: It is not clear here. The authors suggest that the increase of OM during heating was contributed by POA and the secondary formation of emitted VOC? Also, it is interesting to know what could be the reason for the increase of un-resolved (others) during heating? Is it because of the combustion of coal and others during heating which leads to an increase in the proportion of components such as heavy metals? In addition, the large temperature and humidity difference before and after heating may be one of the reasons for the SIA difference, since temperature and humidity affect the partitioning of NO3 and SO4, which can be analyzed more detailedly by the authors in a discussion combined with previous reports.

**1.7.** Line 294: The author thinks that there are only two sources for NAC and OPAH, combustion sources and secondary formation sources, have they been reported before, will there be other sources? According to Cai et al., 2022 and the text above, has biomass burning been included in the combustion source here?

---

## Author Response (AR1)

**Responses to the comments from reviewer #1**

**General comments:**

The manuscript focuses on understanding the sources and contributions of brown carbon (BrC) in urban Beijing. The study was conducted during autumn and winter 2017-2018, analyzing nitrated aromatic compounds (NACs) and oxygenated derivatives of polycyclic aromatic hydrocarbons (OPAHs) in $PM_{2.5}$. It identifies significant sources of 9 NACs and 8 OPAHs in before and during the heating periods in Beijing and highlights the role of secondary formation processes in contributing to BrC, especially during pollution events. The authors conclude that both OPAHs and NACs concentrations were considerably higher during the heating period compared to the pre-heating phase. It was observed that photochemical reactions primarily drove the secondary formation of these compounds before the heating season, whereas aqueous processes were predominant during the heating period. This paper is within the scope of ACP and might be of great interest to the broad atmospheric science community. However, there are areas for improvement in terms of clarity and depth in the discussion of certain results, particularly the implications of the findings for broader atmospheric science and policymaking. Currently, it tends to read more as a data-centric measurement report, making it challenging to connect the introduction and discussion sections logically. Addressing these issues will enhance the manuscript's coherence and relevance. I have a few specific questions and comments that should be addressed before the manuscript can be considered for publication. Language editing is also recommended to refine the presentation of the content.

**Response:** Suggestion taken. We have responded to the specific questions and comments point by point, and asked professionals to edit the language. Detailed responses can be found in the information below.

**Specific comments:**

**1.** Can you provide an estimation of the proportion of the 9 NACs and 8 OPAHs measured in this study relative to the total atmospheric NACs and OPAHs? Additionally, how significant is their contribution to the overall BrC concentration?

**Response:** Research on organic aerosols at the molecular level has been very limited, especially for brown carbon (BrC). To our knowledge, it is currently not possible to quantify the total amount of NACs and OPAHs and the overall BrC in the atmosphere. It is mainly based on existing authentic standard to measure specific species in the atmosphere. Similar to this work, previous studies of their specific composition have also mainly focused on these species (Li et al., 2020a; Wu et al., 2020; Li et al., 2020b; Lu et al., 2019b; Lu et al., 2019a).

**2.** Line 158: The rationale for selecting the lowest 15% as representative of dominant

primary sources for NACs and OPAHs is not clear. Could you provide more detailed reasoning or a sensitivity analysis to support this choice?

**Response:** In this study, the contributions of secondary oxidation to the detected NACs and OPAHs were evaluated by using a CO-tracer method, which is comparable to the EC-tracer used for secondary OC quantification. Various methodologies have been similarly adopted successfully in other studies (Liu et al., 2023; Cai et al., 2022; Chen et al., 2022; Li et al., 2019; Salvador et al., 2021). ([NACs] /[CO]) $_{pri}$ or ([OPAHs] /[CO]) $_{pri}$ represents the primary emission ratio of NACs or OPAHs in relation to combustion. This calculation assumes that the primary source was predominant during the period, with minimal secondary production. However, the primary ratio varies with time, so a fixed primary ratio results in large uncertainties. In addition, secondary compounds may contribute to ambient particles during the selected primary compounds-dominated periods (Zhang et al., 2015). Therefore, in order to minimize deviations, we draw on the work of some well-known scholars to calculate this ratio, using the average of the minimum 15% ratios ([NACs]$_t$ /[CO] or [OPAHs] /[CO]) in the whole campaign, and obtain an acceptable results (Chen et al., 2022; Liu et al., 2023).

**3.** Line 289: Please include the correlation coefficient (r-value) for the pre-heating period in the text. The table shows an r-value of 0.210, which doesn't appear much lower than that of the heating period. Therefore, the statement regarding the significantly higher correlation of NACs with SO$_2$ during the heating period (r=0.275, p<0.05) compared to pre-heating should be cautiously interpreted.

**Response:** Suggestion taken. We have included the correlation coefficient (r-value) for the pre-heating period in the text. See line 309. Numerically, 0.21 doesn't appear much lower than that of the heating period (0.275). But statistically, the *p* value was higher than 0.05 before heating without reaching significance, and it was lower than 0.05 during heating (Table 2). Therefore, the results indicated that coal combustions play a more significant role in NAC formation after heating commences.

**4.** Line 332: Referring to SO$_2$ as an inert chemical may be misleading, as its atmospheric chemical lifetime ranges from days to weeks.

**Response:** Suggestion taken. We have corrected the relevant expression. See line 364-369.

**5.** Lines 341-343: The conclusion that "photochemical oxidation before heating and aqueous oxidation during heating are important" is primarily based on correlations between OPAHs, NACs, O$_3$, or RH. Could you provide additional analysis or reasoning to substantiate this conclusion? Also, please address the consistency of the negative correlation between OPAHs and NACs with O$_3$ during the heating period with the proposed aqueous oxidation processes.

**Response:** Suggestion taken. We have combined the suggestions of multiple reviewers and added some evidences from single particle analysis to well prove the aqueous formation of secondary BrC. "Furthermore, spherical primary OM particles (i.e.,

tarballs), which are mainly from residential coal burning especially during heating period in the North China Plain, usually contain BrC species and their aqueous formation could occur during the long-range transport (Zhang et al., 2021; Zhang et al., 2023)". See line 359-363. Moreover, the negative correlation between NACs and OPAHs with the $O_3$ may be related to the meteorological conditions during the sampling period, e.g. lower temperature and weak solar irradiation during heating period, which suppress the photo-degradation process of NACs and OPAHs. (Li et al., 2020a), and we have added the discussion in line 322-325.

**Technical corrections:**

**1.** Line 76: Please include the publication year in the citation "Ma et al"
**Response:** Suggestion taken.

**2.** Line 90: Please include the publication year in the citation "Huang et al."
**Response:** Suggestion taken.

**3.** Line 108: Consider changing "2017/2018" to "2017 and 2018" for clarity.
**Response:** Suggestion taken.

**4.** Figure 4: The resolution of this figure is notably low; enhancing it would improve clarity.
**Response:** Suggestion taken. We have re-uploaded the Figure.

**5.** Line 331: It would be beneficial to define the specifics of the box plots and error bars
**Response:** Suggestion taken. We have defined the specifics of the box plots and error bars for Figures 3, 5, and 6.

**Responses to the comments from reviewer #2**

**Major Comment:**

In view of its unique light absorption properties and photochemical reactivity, brown carbon is believed to have potential impacts on global climate and atmospheric environmental quality. Therefore, brown carbon has become a hot topic in the field of environmental research in recent years. This paper uses offline methods and GC/MS to investigate the concentration variation and source characteristics of various brown carbon molecules before and during heating period in Beijing. It is found that vehicle emissions, biomass burning and coat combustion are important sources of brown carbon before and during heating period. And secondary source was important for brown carbon. However, the information of the dozen or so molecules that serve as BrC are lacking and the source apportionment are needed more logically accurate and indepth analysis. Here are the comments.

**1.** The title of this article suggests that the research object is brown carbon, and the author has surfaced in the text that NAC and OPAH are two types of substances that have been confirmed as BrC molecules in previous reports. The problem is that the objects studied by the authors are actually more than a dozen confirmed molecules, some of which have been studied and confirmed as BrC molecules, determining the light absorption characteristics of these molecules, such as xie et al., 2017 and Huang, R.-J., 2018. In order to make the conclusions of the paper more rigorous, the author should supplement relevant information on the light absorption parameters of these molecules, otherwise BrC and these more than a dozen molecules are not equivalent.

**Response:** Our group has been looking at the optical properties of related species and has published relevant researches in recent years (Liu et al., 2023; Wu et al., 2020; Li et al., 2020a). As suggested by the reviewers, we have added some discussion of the optical properties of these species. See line 96-101.

As the reviewer said, brown carbon is one of the most important light absorbing aerosols in the atmosphere, and the molecular structure determines the light absorption strength of brown carbon. The light absorption characteristics and formation and transformation mechanism of brown carbon have been the focus and hot spot in the field of atmospheric research. In order to provide recommendations for the prevention and control of pollution events that occurred during this period, this study does not focus on the light absorption characteristics of these species, but focuses on the effects of heating on their molecular composition and sources, especially the comparative analysis before and during heating. A comparative analysis of the chemical constitution of $PM_{2.5}$ and BrC in four different haze events also revealed that the haze in the autumn and winter was caused by SOC during heating and by nitrate prior to heating. Increased attention should be directed towards reducing the emissions of aromatic hydrocarbons and other anthropogenic VOCs when heating commences. This focus is crucial for effectively mitigating pollution and ensuring the preservation of human health.

**2.** What were the reasons behind the authors' selection of CO over levo as a tracer? In my interpretation, the long-lived and inert nature of the tracer (as per the principle and assumptions of the EC-tracer method) ensures that its mass concentration closely approximates that at the start of emission during atmospheric transport. Consequently, the initial emission value can be derived through an emission ratio calculation, as illustrated by Eq. 1. Both levo and CO satisfy this condition, so it is reasonable to use both as tracers in the calculation. However, the authors state otherwise, citing the literature. This prompted the reviewers to question. The authors should provide a more comprehensive explanation of the fundamental assumptions and principles underlying this method, ensuring that readers can comprehend it thoroughly. The correlation analysis reveals a strong between NAC and OPAH with levo, suggesting a preference towards biomass combustion sources. This finding contradicts the author's initial conclusion. Would the use of levo as a tracer yield a different outcome?

**Response:** Both levoglucosan (lev.) and CO satisfy this condition and can be used as the tracer. Moreover, In fact, when doing this part of work, authors calculated lev. and

OC as tracer respectively, and found that there are some differences between them. See Table 1 below. Another important reason, lev. is more important as a tracer of biomass combustion, and the use of CO as a tracer indicates both biomass combustion and coal burning. At the same time, our results confirm that biomass burning is the significant source of NACs and OPAHs in Beijing's urban area during autumn and winter, and the contribution from coal combustion increased notably with the onset of heating during this period. Thus the authors selected CO as the tracer.

Table 1 Contributions (%) of secondary formation (SNACs, SOPAHs) to the total before and during heating periods, based on CO and lev. as the tracer, respectively.

|  |  | CO as the tracer | lev. as the tracer |
|---|---|---|---|
| SNACs | Before heating | 90 ± 11 (51-98) | 53 ± 22 (4.7-86) |
|  | During heating | 98 ± 2.7 (90-99) | 78 ± 9.6 (42-93) |
| OPAHs | Before heating | 86 ± 12 (46-98) | 47 ± 18 (11-78) |
|  | During heating | 96 ± 4.7 (80-99) | 69 ± 9.9 (49-87) |

**Other comment:**

**1.** Line 25: From the content of the article abstract, it seems to be expressed here during heating rather than after heating.

**Response:** We are sorry to cause some trouble to the reviewer. The authors did analyze the data before and during heating.

**2.** Line 68-70: 67% is a specific number, is it determined for all or for a certain site at a certain time?

**Response:** This number is an average from particular studies for certain sites and certain times (Liu et al., 2018; Song et al., 2018). The authors have revised the representation. See line 68.

**3.** Line 79 Please re-write citation formats correctly.
**Response:** Suggestion taken.

**4.** Line 96: Cai et al., 2022 and Yi Chen et al., 2022 did not study the importance of NAC as brown carbon but studied the NAC itself, please quote the literature correctly.
**Response:** Suggestion taken. The authors have put them in their proper place.

**5.** Line 164: Does this period indicate the whole study period or the 15% point selected by the authors? The assumption that the lowest 15% of the ratio of NAC/CO is used to calculate emission ratios, is there any or some direct data or reports on emissions that can be used to justify that this proportion is correct to some extent?
**Response:** This period indicate the whole study period. ([NACs] /[CO]) $_{pri}$ or ([OPAHs] /[CO]) $_{pri}$ represents the primary emission ratio of NACs or OPAHs in relation to combustion. This calculation assumes that the primary source was predominant during the period, with minimal secondary production. However, the primary ratio varies with

time, so a fixed primary ratio results in large uncertainties. In addition, secondary compounds may contribute to ambient particles during the selected primary compounds-dominated periods (Zhang et al., 2015). Therefore, in order to minimize deviations, we draw on the work of some well-known scholars to calculate this ratio, using the average of the minimum 15% ratios ([NACs]$_t$ /[CO] or [OPAHs] /[CO]) in the whole campaign, and obtain an acceptable results (Chen et al., 2022; Liu et al., 2023).

**6.** Line 209: It is not clear here. The authors suggest that the increase of OM during heating was contributed by POA and the secondary formation of emitted VOC? Also, it is interesting to know what could be the reason for the increase of unresolved (others) during heating? Is it because of the combustion of coal and others during heating which leads to an increase in the proportion of components such as heavy metals? In addition, the large temperature and humidity difference before and after heating may be one of the reasons for the SIA difference, since temperature and humidity affect the partitioning of NO$_3$ and SO$_4$, which can be analyzed more detailedly by the authors in a discussion combined with previous reports.

**Response:** In this study, the changes in the associated pollution gases here can account for the increased contribution of POA and VOCs oxidation, although there is currently no way to accurately measure and calculate POA, and we lack VOCs data for the same period. For example, SO$_2$ concentrations during heating (4.3 ± 1.5 ppb) were more than twice that before heating (2.1 ± 0.8 ppb), which is usually as the tracer for coal burning. Furthermore, the increase in mass concentrations of K$^+$ and Cl$^-$ suggested that there were other burning activities during heating along with the combustion of coal, e.g. burning of biomass (Bai et al., 2023; Li et al., 2022). Thus the changes of these tracer gases can demonstrate an increase in POA. Additionally, OC has both a primary source and a secondary generation, and EC mainly comes from primary sources. The OC/EC ratio increased by 63% from 2.7 ± 3.3 before heating to 4.4 ± 3.7 during heating, which indicated an increase in secondary organic carbon (SOC), and SOC is closely related to the oxidation of VOCs. The other components in PM$_{2.5}$ could be heavy metals or currently unknown species maybe from the combustion of coal and others during heating. And these large percentages of unknown objects have also been found sometimes in previous reports (Wang et al., 2017; Li et al., 2016). As suggested by the reviewers, we also added some discussion about the reasons for the changes of SIA. See line 215-221.

**7.** Line 294: The author thinks that there are only two sources for NAC and OPAH, combustion sources and secondary formation sources, have they been reported before, will there be other sources? According to Cai et al., 2022 and the text above, has biomass burning been included in the combustion source here?

**Response:** Biomass burning is contained in the combustion sources, and the results of this work demonstrate that biomass burning played significant roles in the accumulation of NACs and OPAHs in urban Beijing during autumn and winter. See Section 3.3.

**Responses to the comments from reviewer #3**

**Specific comments:**

This study investigated nitrated aromatic compounds (NACs) and oxygenated derivatives of polycyclic aromatic hydrocarbons (OPAHs) in PM2.5 in urban Beijing during autumn and winter. This study showed that the concentrations of both NACs and OPAHs were approximately 2-3 times higher in the heating period than before heating. This study suggested that the relative molecular composition in NACs did not change significantly before and during heating, but Anthraquinone in OPAHs increased by more than twice. This study further revealed the major sources of these NACs and OPAHs and the formation mechanism of secondary BrC before and during heating. Although this study clearly expressed the results, some problems listed below need to be addressed before the manuscript could be published.

**1.** L166: Why did the authors choose 15% lowest [NACs] $_t$ /[CO] ratios as the ([NACs]$_t$ /[CO]) $_{pri}$? Please provided reference materials. Same problem to OPAHs.

**Response:** ([NACs] /[CO]) $_{pri}$ or ([OPAHs] /[CO]) $_{pri}$ represents the primary emission ratio of NACs or OPAHs in relation to combustion. This calculation assumes that the primary source was predominant during the period, with minimal secondary production. However, the primary ratio varies with time, so a fixed primary ratio results in large uncertainties. In addition, secondary compounds may contribute to ambient particles during the selected primary compounds-dominated periods (Zhang et al., 2015). Therefore, in order to minimize deviations, we draw on the work of some well-known scholars to calculate this ratio, using the average of the minimum 15% ratios ([NACs]$_t$ /[CO] or [OPAHs] /[CO]) in the whole campaign, and obtain an acceptable results (Chen et al., 2022; Liu et al., 2023).

**2.** L197: wrong $NO_3^-$.

**Response:** The authors have revised it.

**3.** Figure 2: The percentages of other components in PM2.5 are very large. This result is different from other studies. What are the other components? What causes so many other components?

**Response:** The other components in PM2.5 could be heavy metals or currently unknown species maybe from the combustion of coal and others during heating. And these large percentages of unknown objects have also been found sometimes in previous reports (Wang et al., 2017; Li et al., 2016).

**4.** L338-340: The cited references all used the bulk methods to analyze the formation of secondary BrC. How about single particle analysis? The authors should add some evidences from single particle analysis to well prove the aqueous formation of secondary BrC (Zhang et al., 2023; Zhang et al., 2021).

**Response:** Suggestion taken. The authors have carefully studied the articles

recommended by the reviewer and added some evidences from single particle analysis to well prove the aqueous formation of secondary BrC. "Furthermore, spherical primary OM particles (i.e., tarballs), which are mainly from residential coal burning especially during heating period in the North China Plain, usually contain BrC species and their aqueous formation could occur during the long-range transport (Zhang et al., 2021; Zhang et al., 2023)". See line 359-363.

**Reference list:**

[revised manuscript text omitted]